# Overuse-Related Injuries of the Musculoskeletal System: Systematic Review and Quantitative Synthesis of Injuries, Locations, Risk Factors and Assessment Techniques

**DOI:** 10.3390/s21072438

**Published:** 2021-04-01

**Authors:** Amaranta Orejel Bustos, Valeria Belluscio, Valentina Camomilla, Leandro Lucangeli, Francesco Rizzo, Tommaso Sciarra, Francesco Martelli, Claudia Giacomozzi

**Affiliations:** 1Interuniversity Centre of Bioengineering of the Human Neuromusculoskeletal System (BOHNES), Department of Movement, Human and Health Sciences, University of Rome “Foro Italico”, 00135 Rome, Italy; amaranta.orejel@gmail.com (A.O.B.); valeria.belluscio@gmail.com (V.B.); valentina.camomilla@uniroma4.it (V.C.); l.lucangeli@studenti.uniroma4.it (L.L.); 2Joint Veterans Defence Center, Army Medical Center, 00184 Rome, Italy; francesco.rizzo@tim.it (F.R.); tommaso.sciarra@esercito.difesa.it (T.S.); 3Department of Cardiovascular and Endocrine-Metabolic Diseases and Aging, Italian National Institute of Health, 00161 Rome, Italy; francesco.martelli@iss.it

**Keywords:** musculoskeletal injury, mechanical overload, stress fracture, muscle fatigue, functional assessment, risk factors, assessment instrumentation, assessment protocol

## Abstract

Overuse-related musculoskeletal injuries mostly affect athletes, especially if involved in preseason conditioning, and military populations; they may also occur, however, when pathological or biological conditions render the musculoskeletal system inadequate to cope with a mechanical load, even if moderate. Within the MOVIDA (Motor function and Vitamin D: toolkit for risk Assessment and prediction) Project, funded by the Italian Ministry of Defence, a systematic review of the literature was conducted to support the development of a transportable toolkit (instrumentation, protocols and reference/risk thresholds) to help characterize the risk of overuse-related musculoskeletal injury. The PRISMA (Preferred Reporting Items for Systematic Reviews and Meta-Analyses) approach was used to analyze Review papers indexed in PubMed and published in the period 2010 to 2020. The search focused on stress (overuse) fracture or injuries, and muscle fatigue in the lower limbs in association with functional (biomechanical) or biological biomarkers. A total of 225 Review papers were retrieved: 115 were found eligible for full text analysis and led to another 141 research papers derived from a second-level search. A total of 183 papers were finally chosen for analysis: 74 were classified as introductory to the topics, 109 were analyzed in depth. Qualitative and, wherever possible, quantitative syntheses were carried out with respect to the literature review process and quality, injury epidemiology (type and location of injuries, and investigated populations), risk factors, assessment techniques and assessment protocols.

## 1. Introduction

Overuse-related injuries, hereby grouped as stress fractures of the bones [1,2,3], stress injuries of the musculoskeletal system [4,5,6], and muscle fatigue due to mechanical overload [7], occur when the involved structures fail to adapt to increased mechanical stress resulting from an actual increase in the magnitude of the applied load, an unsustainable number of repetitive loads, abrupt changes in load administration, deterioration of musculoskeletal condition, or a combination of the above [8,9]. Overuse-related injuries mostly affect athletes [7,10], particularly endurance athletes, athletes involved in preseason conditioning [11], and military populations [4,5,8,9,11,12,13,14,15,16]. In the army context specifically, most injuries occur during basic training of new recruits and in special training contexts (incidence up to ≈7% for men and 21% for women [14]). The implications in terms of patient morbidity, recurrence, reduced military performance, and dropout rates do represent a relevant public health issue and a demand for efficient surveillance, prevention, and treatment plans [8,15,17,18].

Among the various risk factors, those worth mentioning are: the increase in ground reaction force during motion tasks; low/reduced muscle strength, body mass index, bone mineral density and vitamin D levels; lifestyle factors; footwear; training and training environment variables; sex; age; ethnicity; history of previous fractures; and altered morphology of the lower limbs [4,6,9,12,18,19,20,21]. As regards vitamin D, the high incidence of D hypovitaminosis in the global population has gained remarkable interest [22,23,24,25,26], as it has been found even in young people and associated with higher percentages of stress fractures [21,27,28,29] or with alterations of muscle function [28,30,31,32,33]. In military populations, there is a significant predisposition to develop low levels of vitamin D mainly due to low exposure to sunlight and the lack of adequately supplemented food [18,27,34]. The potential interference is still to be clarified between vitamin D levels and physical training [17,28,32,35,36] when the latter is either excessive or too poor or missing due to reduced mobility, as in the case of veterans.

Although there are contrasting data on which specific biomechanical assessments are potentially predictive of actual risks of injury [37], functional (biomechanical) biomarkers are investigated as key indicators of lower limb injuries [37] and stress fractures [38] modulated by fatigue [7].

A research project of relevance for the Army Healthcare System, which addressed open scientific questions and involved consolidated technology and the expertise of a team of authors, was approved and funded by the Italian Ministry of Defence in 2019. The core of the MOVIDA Project (Motor function and Vitamin D: toolkit for risk Assessment and prediction) was the design, set up, validation and application of an innovative instrumental toolkit based on objective assessment instrumentation and protocols, for either indoor or outdoor use in several motor tasks, that integrates indicators of the motor function with biological, clinical, environmental and behavioral factors, with the aim of extrapolating helpful information to predict the risk of musculoskeletal injury due to mechanical stress induced by overuse.

The hereby reported systematic review of the literature was thus conducted to gather information necessary to fulfil the main requirements and activities of the project. The aim of the literature search was the full characterization, wherever possible, of overuse-related injuries and muscle fatigue of the lower limbs in association with functional (biomechanical) biomarkers, in terms of the type and location of injuries, populations, risk factors, suitable measurement instrumentation and protocols for functional assessment and diagnosis.

## 2. Materials and Methods

### 2.1. Literature Search and Review Process

The main topic of the search was globally defined as “overuse-related injuries and muscle fatigue of the lower limbs in association with functional (biomechanical) biomarkers: type and location of injuries, populations, risk factors, assessment instrumentation and assessment protocols”. Overuse-related injuries were classified either as stress fractures, referring to fatigue-induced bone fractures caused by repeated submaximal loading, or as stress injuries, the latter being a synthetic way to refer to stress-related injuries of soft tissues such as muscles, tendons and ligaments following their strain under repeated mechanical stress. Neither stress fractures nor stress injuries include acute traumas. The search was carried out within PubMed and constrained to the period of January 2010 to March 2020. Only Review papers on humans were included. Papers were retrieved in three separate searches and merged afterwards: (i) search on stress fractures: keywords were (“stress fracture/s” or “overuse fracture/s” or “overload fracture/s”) and (“lower limb” or “ankle” or “foot”); (ii) search on stress injuries: keywords were (“stress injury/ies” or “overuse injury/ies” or “overload injury/ies” or “stress trauma/s” or “overuse trauma/s” or “overload trauma/s”) and (“lower limb” or “ankle” or “foot” or “trunk” or “back” or “lower back” or “lumbar spine”); (iii) search on muscle fatigue: keywords were “muscle fatigue” and (“lower limb” or “ankle” or “foot” or “leg” or “trunk” or “back” or “lower back” or “lumbar spine”). The authors followed the PRISMA (Preferred Reporting Items for Systematic reviews and Meta-Analyses) guidelines to design the review [39]. The abstract of each publication was independently reviewed by two authors to decide on its relevance for the review; in case of disagreement, a team discussion would ensue. The authors who had no previous experience in this task were trained on similar articles (not included in the search) and paired up with an experienced reviewer. Criteria for exclusion were: not in English, not relevant to the main topic of the search, and full text not retrievable. In detail, for a paper to be relevant to the main topic of the search, inclusion criteria followed in a first-level search were: focus on overuse-related injuries, either in terms of fractures or soft tissue injuries, or muscle fatigue; focus on lower limb injuries, even if in combination with injuries at other body segments; being a review, either narrative, or structured, or systematic, with or without meta-analysis—this last point was mandatory since, in some cases, papers classified and retrieved as reviews do actually consist of other types of publications. Each full text was assigned to one reviewer who discussed eventual concerns with the team. Reviews with detailed information were analyzed in function of the main topics of the search. For those reviews with only marginal or qualitative information, a second-level search was carried out among their bibliographical references. After removing duplicates and checking for exclusion criteria, the added set of articles followed the same review process as for the review papers. Only the inclusion criterion of being a review was removed at this stage of the process. On the basis of the information retrieved and the specific items of interest, the review database was designed to allow the team to carry out qualitative or quantitative syntheses on the type and quality of the reviews, overview of findings, epidemiology, risk factors, assessment instrumentation, and assessment protocols.

### 2.2. Epidemiology

Several epidemiological perspectives can be adopted to summarize reported injuries, and to use these results to inform subsequent actions. Based on the main requirements and activities of the project, information was analyzed for age groups, sex, populations, type of physical activity potentially causing the injury under analysis, injury prevalence, and injury location. As for the type of injury, the study focused on stress fractures, stress injuries, and fatigue. The epidemiological results were therefore categorized according to these three types.

### 2.3. Risk Factors

A risk factor, either intrinsic or extrinsic, is something that makes a person more likely to develop a disease or condition. Intrinsic factors are a person’s individual characteristics (i.e., age); extrinsic factors are external to the individual and include variables such as the type of sport, degree of exposure to a sport, training, and environment. Intrinsic factors can be further categorized as modifiable, potentially modifiable, and nonmodifiable. An intrinsic factor is modifiable when it can be changed as a result of an intervention (i.e., nutrition), nonmodifiable when it cannot be altered (i.e., age), and potentially modifiable when it can be adjusted in specific situations but may not be modifiable in others (i.e., status of bone mineral density). Likewise, extrinsic factors can be categorized as modifiable (i.e., use of adequate shoes) and potentially modifiable (i.e., type of training and training environment). In this review, intrinsic and extrinsic risk factors were categorized as modifiable, nonmodifiable and potentially modifiable following the criteria proposed by [14].

### 2.4. Assessment Instrumentation and Protocols

Another relevant research question of the MOVIDA project and, consequently, of this literature review, concerned the investigation of the most common instruments and tools used to functionally assess stress fractures, stress injuries, and fatigue. In the case of stress fractures and stress injuries, assessment is intended as either direct—through clinical examination and/or diagnostic instrumentation—or indirect (functional), the latter based on the assessment of the impact of stress damage on musculoskeletal performance during assigned motor tasks. As for muscle fatigue due to mechanical overload, the search focused on functional assessment only.

Assessment may be further grouped into objective and subjective assessment, the latter including assessment tools traditionally used in occupational ergonomics (among which self-administered questionnaires, qualitative or semi-quantitative, operator-mediated scales, psycho-physical and psycho-social scales). Given the focus of the project on an innovative toolkit to perform objective measures, we limited our analysis to objective assessment, requiring measurement devices.

Four main categories of objective assessment instrumentation were identified to address the analysis of the literature, namely, imaging techniques, wearable instrumentation for motion analysis, laboratory (i.e., non-wearable) devices for motion analysis, and instruments for physiological assessment.

Protocols contemplating imaging techniques were not investigated; rather, functional assessment protocols based on technology and, when retrieved, protocols to evaluate the vitamin D level or bone density were separately synthesized for stress fractures, stress injuries and fatigue.

## 3. Results

### 3.1. Summary of the Literature Review Process

The flowchart of the review process following PRISMA guidelines for systematic reviews is reported in Figure 1. The literature search of reviews resulted in 225 papers; 14 of them were excluded because they were not in English, plus another 96 because they were found not to be relevant in the abstract review process. After full-text analysis of the remaining 115 papers, 17 were discarded because they were not relevant. References of poorly informative review papers were explored (second-level search) and reviewed following the same procedures: they produced an additional set of 141 papers —reviews and original research papers—85 of which were included. From the total of 183 selected papers, the quality assessment process led to the exclusion of 74 papers containing only marginal or introductory information; those papers, though discussing aspects of the search main topic, either focused on different research questions or only reported qualitative comments or too general information. This subset of papers is briefly accounted for in Section 3.2.

The remaining 109 papers [2,7,8,14,19,27,38,40,41,42,43,44,45,46,47,48,49,50,51,52,53,54,55,56,57,58,59,60,61,62,63,64,65,66,67,68,69,70,71,72,73,74,75,76,77,78,79,80,81,82,83,84,85,86,87,88,89,90,91,92,93,94,95,96,97,98,99,100,101,102,103,104,105,106,107,108,109,110,111,112,113,114,115,116,117,118,119,120,121,122,123,124,125,126,127,128,129,130,131,132,133,134,135,136,137,138,139,140,141] were judged to be relevant to the review main topic, with detailed information on one or more of the search items as in Figure 1 (PRISMA scheme, at the bottom). They are hereby briefly described:
-A total of 16 were systematic reviews, seven of which were based on PRISMA Statement [39]. All systematic reviews explicitly reported on the quality assessment of selected papers: five of them adopted a peculiar quality assessment based on purposely defined inclusion/exclusion criteria; two used the quality index [142], and each of the remaining nine applied a different quality assessment tool [143,144,145,146,147,148,149,150,151]. Meta-analysis was found in four reviews only. The level of evidence was only quantified in two reviews, level 6 and level 4. The systematic reviews were informative for the items of epidemiology and risk factors; however, they scarcely informed on assessment instrumentation (very few of them dealt with laboratory instrumentation and none with wearable instrumentation for motion analysis) and assessment protocols.-A total of 51 were narrative reviews; two of them showed a level of evidence 5, and one a level of evidence 4; besides contributing to epidemiology and risk factors, 11 contained relevant information for assessment instrumentation for motion analysis; however, just six of them detailed assessment parameters, and only six detailed assessment protocols.-The remaining 42 papers were: 20 interventional studies; eight cohort studies, two of which with a level of evidence 2; two cross-sectional studies; 10 research studies; one commentary (relevant since highly informative with respect to one of the included systematic reviews); one paper on SENIAM guidelines on surface electromyography (EMG) (relevant and informative for the item of assessment protocols). With respect to assessment issues, 33 out of those 42 papers contributed to the assessment instrumentation for motion analysis (either laboratory or wearable instrumentation), 29 detailed biomechanical and physiological assessment parameters, 35 contributed assessment protocols.

### 3.2. Qualitative Synthesis of Marginal/Introductory Papers

The 74 marginal/introductory papers [152,153,154,155,156,157,158,159,160,161,162,163,164,165,166,167,168,169,170,171,172,173,174,175,176,177,178,179,180,181,182,183,184,185,186,187,188,189,190,191,192,193,194,195,196,197,198,199,200,201,202,203,204,205,206,207,208,209,210,211,212,213,214,215,216,217,218,219,220,221,222,223,224,225] were 40 reviews (either systematic or narrative) dealing with stress fractures or injuries, 15 reviews addressing muscle fatigue, and 19 papers (clinical or research studies, or case reports) obtained through the second-level search and mentioning one or more items of this literature search. Many of those papers focused on the clinical diagnosis and treatment of overuse-related injuries. Although they delivered no detailed quantitative information on this review’s search items, they did mention or briefly comment on them, thus strengthening some evidence on:-The occurrence of stress fractures: mentioned in 34 papers, they were associated with lower limb (18), foot (15, mostly metatarsals), pelvis (8), spine (8), ankle (7), trunk (3) and upper limb (3).-The occurrence of stress injuries: mentioned in 36 papers, they were associated with lower limb (27), foot (10), ankle (10), upper limb (7), pelvis (5), trunk (5) and spine (4).-The occurrence of overuse-related muscle fatigue: mentioned in 19 papers, fatigue was associated with trunk (7), lower limb (6), upper limb (3), foot (1) and pelvis (1).-The higher risk of injury experienced by elite athletes (34), recreational athletes (18) and soldiers (14).-The risk of injury associated with running (18), military activities/training (14), walking and march (six in total).-Risk factors: mentioned in 44 papers, 10 of which without any detail or just mentioning them as intrinsic or extrinsic factors; explicitly mentioned risk factors were training variables also including the type and environment (19 in total), altered biomechanics of motion (10), systemic or other diseases (8), morphology and anatomical factors (7), sex (6), body composition (6), alteration of bone density or vitamin D level (6), age (5), previous stress fractures (5), footwear (3) and ethnicity (1).-Assessment instrumentation: mentioned in 39 papers, it mostly refers to imaging (25); instrumentation for biomechanical assessment is also mentioned (6, 3 of which about EMG).-Assessment protocols: mentioned in 26 papers, 15 of which mentioning physical or clinical examination (one also referring to vitamin D level and bone density) and nine citing functional tests (eight also citing biomechanical parameters).

### 3.3. Epidemiology of Overuse-Related Injuries

The literature search produced a large amount of data concerning the epidemiology of the included studies [2,7,8,14,19,27,38,40,41,42,43,44,45,46,47,48,49,50,51,52,53,54,55,56,57,58,59,60,61,62,63,64,65,66,67,68,69,70,71,72,73,74,75,76,77,78,79,80,81,82,83,84,85,86,87,88,89,90,91,92,93,94,95,96,97,98,99,100,101,102,103,104,105,106,107,108,109,110,111,112,113,114,115,116,117,118,119,120,121,122,123,124,125,126,127,128,129,130,131,132,133,134,135,136,137,138,139,140,141]. Figure 2 gives a visual representation of the main information extrapolated from the literature, concerning age (age-group division according to [226]), sex, investigated populations, type of physical activity, prevalence and location of injuries. The majority of studies focused on adult people (19 to 65 years), the most investigated population being professional athletes (>6 h/week of physical activity), followed by recreational athletes and militaries. As for the type of physical activity, 50% of the revised studies focused on running, while for the type of injury, the most reported were stress fractures and stress injuries, also associated with the site of injury (here grouped as lower limbs, pelvis, spine, and upper limbs) and the type of activity; less details were found for fatigue.

Injury locations are reported in Figure 3 for stress fractures, stress injuries, and muscular fatigue. The figure shows that the foot is the most reported body site (30 articles for stress fractures, 15 articles for stress injuries), while the spine is the most reported for fatigue, other than lower limb fatigue, which was the main focus of the string search.

Articles provided further detailed information about the body location of stress fractures, which are summarized in Figure 4 (upper limb, trunk, spine and pelvis) and Figure 5 (lower limb, ankle and foot). One of the most common locations of stress fractures in the pelvis is the sacrum, followed by ischiopubic ramus, often in the inferior pubic ramus adjacent to the symphysis (Figure 4b). When referring to lower limbs (Figure 5a), the most frequent injury occurs at the tibia, but several studies also reported stress fractures of the femoral neck and femoral shaft. The medial malleolus is the most common location of stress fracture at the ankle level (Figure 5b), while the metatarsal heads are the most common sites of injury at the foot level, as well as calcaneal stress fractures and tarsal navicular (Figure 5c). For the sake of clarity, it is worth reminding here that the search strategy of this review did not explicitly include upper limb overuse-related injuries; however, several papers also reported detailed information with respect to that body area, which have been reported for completeness. Of course, for that body segment, the search was neither systematic nor exhaustive.

### 3.4. Risk Factors

The following main risk factors were retrieved from the literature [2,7,8,14,19,27,38,40,41,42,43,44,45,46,47,48,49,50,51,52,53,54,55,56,57,58,59,60,61,62,63,64,65,66,67,68,69,70,71,72,73,74,75,76,77,78,79,80,81,82,83,84,85,86,87,88,89,90,91,92,93,94,95,96,97,98,99,100,101,102,103,104,105,106,107,108,109,110,111,112,113,114,115,116,117,118,119,120,121,122,123,124,125,126,127,128,129,130,131,132,133,134,135,136,137,138,139,140,141], sorted into categories and listed in line with their occurrence, in decreasing order, as follows:-Intrinsic modifiable factors: biomechanical movement patterns, entry-level of physical fitness, body composition, other health risk behaviors, and nutritional factors;-Intrinsic nonmodifiable factors: sex, morphology and anatomical factors, age, history of lower limb stress fractures, and ethnicity;-Intrinsic potentially modifiable factors: bone mineral density;-Extrinsic modifiable factors: training volume, training intensity, footwear, training duration, training frequency, distance, and all training variables;-Extrinsic potentially modifiable factors: type of training and training environment.

Figure 6 summarizes the number of articles for each category and subcategory of risk factors.

Factors were also synthesized with respect to the main populations found in the literature: military, professional athletes, recreational athletes, and sedentary. Figure 7 and Figure 8 address intrinsic and extrinsic factors, respectively.

### 3.5. Objective Assessment Instrumentation

Instrumentation can be either used in direct assessment, as a diagnostic tool, or for indirect (functional) evaluations of the impact of stress damage on musculoskeletal performance during assigned motor tasks. In the case of stress fractures and stress injuries, direct assessment usually includes clinical examination, which precedes referral to imaging diagnostic analyses since, along with the patient’s history, it provides the foundation for making a diagnosis of stress fracture.

Twenty-one papers included a detailed description of the physical examination performed for specific stress fractures [8,14,46,52,56,65,68,77,78,83,85,87,91,98,101,107,118,119,120,126,217]. They recall the highly suggesting role of the patient’s history and symptom description, and the confirmatory role of physical examination. Detail is provided on how to identify areas of tenderness and swelling through palpation and, less often, how to reproduce symptoms through specific clinical tests.

As regards the instruments involved in quantitative assessments, four main categories came out from the literature search [2,7,8,14,19,27,38,40,41,42,43,44,45,46,47,48,49,50,51,52,53,54,55,56,57,58,59,60,61,62,63,64,65,66,67,68,69,70,71,72,73,74,75,76,77,78,79,80,81,82,83,84,85,86,87,88,89,90,91,92,93,94,95,96,97,98,99,100,101,102,103,104,105,106,107,108,109,110,111,112,113,114,115,116,117,118,119,120,121,122,123,124,125,126,127,128,129,130,131,132,133,134,135,136,137,138,139,140,141], covering the domains of imaging techniques, wearables and laboratory devices, and instruments for physiological assessment.

The literature search revealed that imaging was the most used technology (33 articles), followed by laboratory devices for motion analysis (26 articles), wearable instrumentation for motion analysis (25 articles), and instruments to monitor physiological parameters (10 articles). Few papers proposed a more complete assessment based on a combined approach of the mentioned groups of instruments, as reported in Table 1.

From the performed literature search, X-rays, MRI, and CT scan were the most used imaging techniques (Figure 9).

Figure 10 shows the occurrence of laboratory and wearable instrumentation, and instrumentation for physiological assessment in the retrieved selected articles. According to these findings, the functional assessment of overuse-related injuries mainly relies on laboratory optoelectronic stereophotogrammetric systems, followed by force platforms, while the most used wearable devices were inertial measurement units (IMUs) and electromyographic systems (EMG). The most used devices for physiological assessment monitored the heart rate, lactate concentrations, and changes in VO2 max, in that order.

The use of each of the four above-mentioned groups of measurement techniques has been further analyzed for stress fractures, stress injuries, and fatigue (Figure 11). No technique displayed any prevalence with respect to the type of injury aside from the obvious prevalence of imaging techniques in relation to fractures.

### 3.6. Assessment Protocols

The literature review provided several assessment protocols associated with overuse-related injuries [2,7,8,14,19,27,38,40,41,42,43,44,45,46,47,48,49,50,51,52,53,54,55,56,57,58,59,60,61,62,63,64,65,66,67,68,69,70,71,72,73,74,75,76,77,78,79,80,81,82,83,84,85,86,87,88,89,90,91,92,93,94,95,96,97,98,99,100,101,102,103,104,105,106,107,108,109,110,111,112,113,114,115,116,117,118,119,120,121,122,123,124,125,126,127,128,129,130,131,132,133,134,135,136,137,138,139,140,141]. Besides protocols to evaluate stress fractures, stress injuries, and fatigue, protocols were also found for the specific investigation into running injury, the impact of overload in running, and the impact of overload in load carriage (Figure 12). Five studies were specifically developed to identify signs of muscular fatigue in EMG signals; although all the tests are worth mentioning [43,64,88,99,141], these were not included in the summary of Figure 12 since they are not directly related to overuse-related injuries.

### 3.7. The Occupational Ergonomics Domain

The literature search strategy implemented in this study was not designed to include exhaustive information on other important aspects associated with the topic of overuse-related injuries, as is the case for occupational ergonomics. The investigation of this specific domain was beyond the scope of the study, and the search terms “occupational ergonomics” were not explicitly used. The few findings that came out from the literature database are here shortly summarized without any claim of completeness. Briefly:
-20 papers (nine reviews and 11 original studies) dealt with occupational ergonomics and overuse-related injuries. Specifically, nine referred to the military field (six reviews, three original studies) [67,89,113,127,135,137,170,173,223]; one review dealt with bricklayers and construction supervisors [224]; four (two reviews, two original studies) more generally referred to long-standing posture at work [62,70,133,171]; four (original studies) addressed athletes [102,116,124,209]; two original papers dealt with slip-induced falls in workers [100,114].-Of the 20 papers, five (three reviews, two original studies) dealt with assessment instruments such as the strain index, rapid entire body assessment (REBA) score, pain/discomfort/complaints scales including the VAS, Borg and Likert scale [62,70,133,165,171]; 10 papers (one review, nine original studies) reported models and techniques (regressive/logistic models, odds ratio, hazard ratio) to detect risk factors [46,81,89,102,106,116,121,124,135,173].

## 4. Discussion

### 4.1. The Literature Review Process

This systematic review focused on the investigation of a main topic defined as “overuse-related injuries and muscle fatigue of the lower limb in association with functional (biomechanical) biomarkers: type and location of injuries, populations, risk factors, objective assessment instrumentation and protocols”. Search terms were of course quite recurrent in the scientific literature. For this reason, rather narrow inclusion criteria were defined for the search in addition to the commonly used criteria (papers in English, no grey literature, full text availability), which are hereby briefly motivated:
-Only review papers were included: systematic reviews, besides a comprehensive, critical and updated analysis of the state-of-the-art developments on a research question, contain quality procedures, either standardized or adapted or purposely defined, for the assessment of the quality and adequacy of retrieved information [227]; since the focus on systematic reviews alone would have been too restricting, the search also included reviews that were not structured as systematic.-The search was limited to PubMed: in agreement with the previous limitation, this criterion likely allowed to reach all relevant scientific reviews and represented an acceptable trade-off between depth of the search and resources needed for the search, analysis and synthesis.-The search was limited to the last decade: this criterion is a consequence of the first one, since older evidence or relevant outcomes are surely accounted for in the review papers published in the last decade. Additionally, older publications might not be informative enough about more recent technology, training techniques or clinical approaches for the diagnosis and treatment of musculoskeletal injuries; last, of course, this criterion contributed to optimizing the available resources.

During the abstract screening of the retrieved 211 reviews (225 minus 14 excluded because not in English), the need clearly emerged to deepen the search into assessment instrumentation and protocols, since information about functional assessment (which was the core of the MOVIDA project) seemed inadequate for a comprehensive synthesis to include in this review and to support the project’s main activities. This led to widening the search by including papers cited within those reviews that mentioned the search topics without giving enough details.

At the end of the overall screening, the quality assessment and analysis process of the retrieved papers, 109 papers were selected [2,7,8,14,19,27,38,40,41,42,43,44,45,46,47,48,49,50,51,52,53,54,55,56,57,58,59,60,61,62,63,64,65,66,67,68,69,70,71,72,73,74,75,76,77,78,79,80,81,82,83,84,85,86,87,88,89,90,91,92,93,94,95,96,97,98,99,100,101,102,103,104,105,106,107,108,109,110,111,112,113,114,115,116,117,118,119,120,121,122,123,124,125,126,127,128,129,130,131,132,133,134,135,136,137,138,139,140,141] which adequately addressed all the search topics and allowed for the quantitative syntheses reported in the results and briefly discussed below. Another 74 papers [152,153,154,155,156,157,158,159,160,161,162,163,164,165,166,167,168,169,170,171,172,173,174,175,176,177,178,179,180,181,182,183,184,185,186,187,188,189,190,191,192,193,194,195,196,197,198,199,200,201,202,203,204,205,206,207,208,209,210,211,212,213,214,215,216,217,218,219,220,221,222,223,224,225] with a focus on different research questions or only qualitatively commenting on the search topics were classified as marginal or introductory, and were used to carry out a qualitative synthesis (summarized in Section 3.2) which strengthened the relevance of, and the scientific interest towards, overuse-related injuries, and the lack of thorough assessment models to account for the multidimensional set of risk factors.

### 4.2. Epidemiology of Overuse-Related Injuries

It became clear that the majority of studies focused on adults (19 to 65 years), while subjects of pediatric age are the least investigated population. Some kinds of injury seem to be sustained mostly by women. Regarding the type of physical activity, 50% of the reviewed studies focused on running, whilst the most investigated population is that of professional athletes. There is a significant amount of literature about military populations.

The injuries considered as a design factor of this review were evenly distributed in the selected papers, with a slight prevalence of stress fractures (45% of the papers) over stress injuries (30%), and fatigue (25%). Injury sites vary according to activities; thus, injury location was separately investigated for the three injury categories [75,121,228].

In particular, lower limbs were the most affected location of trauma (>40 articles for fractures, >30 for stress injuries, >20 for fatigue), followed by feet (almost 30 articles for fractures and 15 for stress injuries) and the spine for muscular fatigue (Figure 3). When referring to lower limbs (Figure 5a), several studies reported stress fractures of the femoral neck and femoral shaft, but the tibia, and more precisely its medial border, resulted as the most frequently injured bone of the lower limb; medial tibial stress syndrome (MTSS) was reported as one of the most common causes of exercise-induced lower leg pain, with incidence reaching 20% in runners and 35% in military personnel [46]. As a key element of the kinetic chain of the human locomotor system, the tibia plays a connecting function between the foot–ankle complex, which directly manages the body-ground interaction, and the hip joint, which transmits upper body motion to the bottom elements of the chain. Thus, abnormalities in the structure or function of either the upper-tibia or lower tibia segments contribute to the tibia burden in managing repetitive loading cycles, especially during the landing and propulsion phases of gait or running. Among the MTSS risk factors, with strong or moderate evidence, in fact, the literature includes hip external-rotation ROM, ankle plantar-flexion ROM, and navicular drop, besides increased BMI either due to increased body weight or to external load carriage [42,46,63,75,103,104,116,121,130]. Correspondingly, the medial malleolus, which is the most distal part of the tibia, resulted as the most common location of stress fracture at the ankle level (Figure 5b). At the foot level, metatarsals from the second to fifth, which are the most involved foot structures during propulsion phases and, for some runners, also act as dampers during forefoot landing, resulted as the most common sites of injury. Calcaneal and tarsal navicular stress fractures were frequently found as well (Figure 5c), reasonably associated with the relevant action of those structures in impact absorption during foot landing.

As clarified in the Results section, Section 3.3, the search strategy of this review did not explicitly include upper limbs, but since overuse-related injuries to this body segment were found in some papers, they have been reported to provide a more complete picture. In any case, no interpretation is hereby given for these data.

### 4.3. Risk Factors

A rigorous evidence-based approach could not be applied to risk factors. Thus, attention was paid to their occurrences in the retrieved literature, as how many times a risk factor had been mentioned within the search database might be accounted for as a possible “proxy” of a relevance/attention indicator.

From a total of 109 articles, intrinsic risk factors (modifiable (83 articles), nonmodifiable (63 articles) and potentially modifiable (20 articles)) emerged with more evidence than extrinsic factors (modifiable (50 articles) and potentially modifiable (45 articles)), with attention paid, in decreasing order, to: altered biomechanics of motion (modifiable), sex (nonmodifiable), morphological and anatomical alteration of the musculoskeletal structures (nonmodifiable), entry level of physical fitness (modifiable), nutrition (modifiable), bone mineral density (potentially modifiable), age (nonmodifiable), history of lower extremity stress fractures (nonmodifiable), and ethnicity (nonmodifiable). Other intrinsic risk factors were reported in terms of behaviors that entail a risk for health, mainly smoking, almost complete absence of physical activity and, in general, factors accounted for in cardiovascular risk assessment, or associated with diseases (among which metabolic, genetic, degenerative, systemic). In particular, women were frequently reported at a greater risk of overuse-related injuries (mainly stress fractures), a risk that was mostly found in association with the following conditions: young women practicing sports at a high level of intensity and suffering from the female athlete triad (disordered eating, amenorrhea, and osteoporosis); women with musculoskeletal peculiarities with respect to men undergoing the same amount of workload (for example lower bone mineral density at the tibia); and women in pre- or post-menopause, frequently associated with osteopenia and osteoporosis.

While some intrinsic risk factors are nonmodifiable, extrinsic factors are all modifiable or potentially modifiable, which means that modulation actions may be undertaken by modifying the applied workload and/or acting on the whole system carrying the load (i.e., the human system, interfaces, tools and environment). In decreasing order, the following extrinsic factors emerged from this review: the type of training (potentially modifiable), training volume and training intensity (modifiable), training environment (potentially modifiable), footwear (modifiable), training duration and training frequency (modifiable), and distance (modifiable). All training variables as a whole are seldom reported in the literature as a risk factor, but each variable individually is highly recurrent, suggesting that individual modulation based on peculiar needs and conditions may help to reduce the overall risk of injury.

The cross-overview of findings on epidemiology and risk factors yielded an additional observation: a key outcome from papers with meta-analysis is that attention to stress fractures decreased from distal to proximal, likely as a result of the prevalence of locomotion tasks involving interaction with the ground. Acting on intrinsic and extrinsic modifiable factors can modulate, but not eliminate, this greater vulnerability of body structures closer to external solicitations.

### 4.4. Assessment Instrumentation

The literature search brought to light four main categories of objective assessment devices covering the domains of imaging techniques, wearables and laboratory devices, and instruments for physiological assessment.

Imaging was the most investigated technology (33 articles), followed by laboratory devices (26 articles), wearables instruments (25 articles), and instruments to monitor physiological parameters (10 articles). It should also be noted that several papers proposed a combined approach based on the adoption of different technologies.

#### 4.4.1. Imaging

Imaging plays a key role in the diagnosis and management of stress fractures and stress injuries. X-ray findings are usually seen two to eight weeks after the appearance of symptoms, and although the sensitivity of radiography is quite low in the early stages of these injuries [133], it is considered the first approach in the case of a suspected fracture. Radiography findings correlate with the magnetic resonance imaging (MRI) signs of the fracture line, while ultrasound has limited diagnostic value in bone stress fractures, but it is commonly used in detecting muscle edema. Computed tomography (CT) scanning on bone is useful in differentiating conditions that mimic stress fractures, while radionuclide bone scanning is highly sensitive but lacks specificity or the ability to directly visualize fracture lines [68]. The literature search showed that X-rays, MRI, and CT scans are the most investigated techniques (Figure 9).

#### 4.4.2. Wearables and Laboratory Devices

With wearable and laboratory instruments and techniques, an objective evaluation can be made of different parameters related to human movements: compared to non-objective techniques, they furnish more efficient measurements, providing a large amount of reliable information. They can be classified according to two approaches based on either non-wearable or wearable devices. Non-wearable systems require the use of controlled research facilities (laboratories) where the devices are located to capture data while the subject performs the activities in a confined laboratory environment. As Figure 10a shows, optoelectronic stereophotogrammetric systems are the most used technique: widely regarded as the gold standard for motion capture, they are the most accurate technique to track the kinematics of human movement [228]. Additionally, force platforms are very commonly used systems. In contrast, wearable devices can allow for data collection outside the laboratory and capture information about the human movement during the person’s everyday activities. They provide a non-invasive, low-cost method to quantify movement, and are easily portable. The wearable devices most commonly identified in this review were inertial measurement units (IMUs) and electromyographic systems (EMG) (Figure 10b).

As Figure 11 shows, stress fractures were mainly evaluated by using imaging techniques, followed by laboratory (non-wearable) devices, while wearable devices and instruments for physiological assessment were mainly employed for assessing fatigue.

### 4.5. Functional Assessment Protocols

Imaging techniques have been discussed in the previous paragraph in terms of instrumentation. They will be no longer discussed in terms of protocols or parameters since they are already well-characterized in the scientific literature. Instead, attention is paid to functional assessment protocols, both because they still need consolidation with respect to the characterization of overuse-related injuries, and because they represent a crucial issue of the MOVIDA Project which originated this systematic review.

Functional assessment protocols were mostly used to investigate muscle fatigue (Figure 12); they mainly consisted of tests during walking, running, jumping, and tests to quantify isometric strength or isokinetic power. Timing, kinematics and kinetics were the most quantified parameters. All the retrieved protocols—detailed in Section 3.6—relied on validated procedures and can be implemented either in motion analysis laboratories or outdoors by using wearable instrumentation and physiological tools. It is worth mentioning that most of them are reliably applicable to the populations investigated within the MOVIDA Project. However, in the presence of more impaired individuals, it is reasonable to prefer walking-related tests and/or to adapt the more demanding ones. Stress injuries were investigated as well through similar consolidated functional tests. As expected, stress fractures mainly rely on imaging techniques; however, some functional tests have been reported, mainly associated with temporal variables, and the detection of the vitamin D level (25OHD level) has been interestingly added to the assessment protocols (Figure 12). While this biomarker was not retrieved in the two previous sets of protocols, its relevance seems worth investigating also with respect to stress injuries and fatigue.

## 5. Conclusions

The scientific community was found to pay considerable attention to the topic of overuse-related injuries, which are to be intended not only as the result of excessive mechanical overload (either under occasional or repetitive conditions), but also as the result of a mechanical stress level which is not bearable by/excessive for the involved musculoskeletal structures. The reasons for the setup of these risk-of-injury conditions shall be searched at a multidimensional level, since they involve at least biological, clinical and behavioral factors, lifestyle, physical, sport or working workload, biomechanics and environment, musculoskeletal structural and functional features. A comprehensive characterization of epidemiology, risk factors, assessment instrumentation and assessment protocols to investigate overuse-related injuries has been found in the reviewed literature, especially regarding sport activities (running being the most investigated and reported one), at a professional more than at a recreational level, and the training and workload of soldiers. Some critical issues still need to be reliably addressed, including: (i) the design and validation of multidimensional interpretative models, which account for the many identified risk factors; (ii) a wider use of consolidated technology, protocols and parameters for functional assessment, either to be performed under controlled laboratory conditions or in more ecological settings. This instrumental setup should not only be aimed at characterizing/quantifying the presence and impact of the injuries but also to contribute to the early detection and minimization of risk conditions, and as a support to treatment plans; (iii) deeper scientific investigation of other populations at risk, among which the elderly; individuals with, or recovering from, systemic diseases; neurodegenerative diseases or traumas; and pre- and post-menopausal women.

This systematic review focused on the collection, analysis and synthesis of evidence, or at least of reliable information to characterize overuse-related damage to the musculo-skeletal system with special focus on lower limb injuries, most frequently reported risk factors, and suitable configurations of objective assessment scenarios (namely, measurement devices, test protocols, measurable quantities or extrapolated indicators). The implemented literature search strategy partly limited the retrieval of exhaustive information on some other relevant aspects associated with overuse-related injuries as, for example, occupational ergonomics (related findings briefly summarized in Section 3.7). The research question also influenced the way retrieved papers were analyzed and synthesized. In particular, investigations did not go in depth into indicators or models to estimate risk factors, such as hazard or odds ratios and regressive or logistic models; assessment instruments—other than objective measurement devices—typically used in clinical or occupational ergonomics examination; occupational scenarios other than the military or ground-related sports context; populations such as developmental-age individuals, seriously impaired individuals, and the elderly; and upper-body segments. Whenever retrieved, however, those findings were briefly summarized and, although it is out of the specific focus of this review, they may represent a starting point to develop toolkits dedicated to other populations (elderly), body localization (upper-body segments) and occupational settings (other workers).

## Figures and Tables

**Figure 1 sensors-21-02438-f001:**
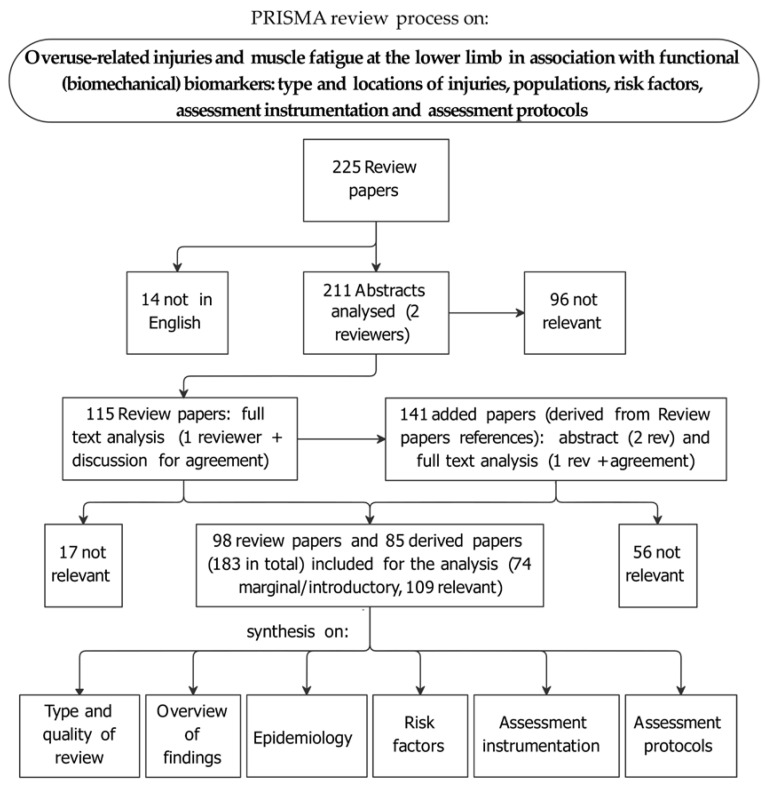
Flowchart of the review process based on PRISMA guidelines for systematic reviews [39].

**Figure 2 sensors-21-02438-f002:**
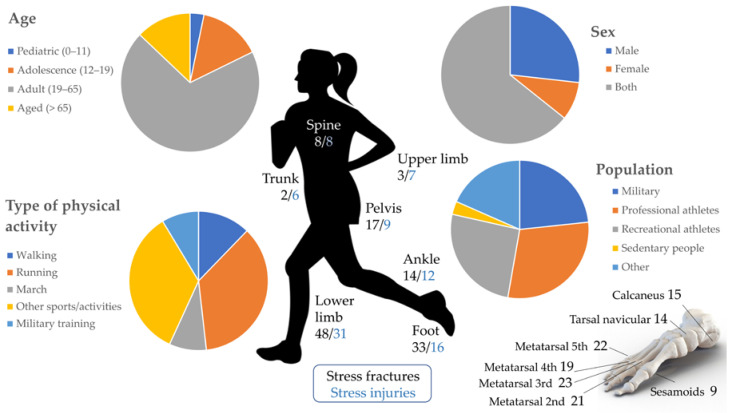
Epidemiology of the reviewed articles regarding age, sex, type of physical activity, population, and location of stress injuries and stress fractures.

**Figure 3 sensors-21-02438-f003:**
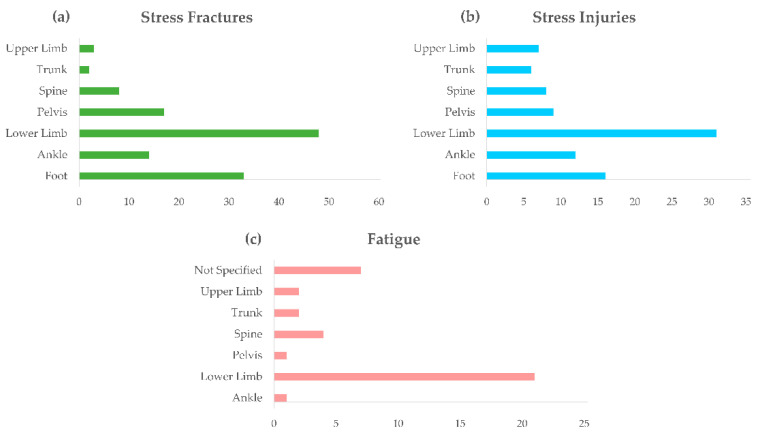
Reviewed articles regarding location of stress fractures (**a**), stress injuries (**b**), and muscular fatigue (**c**).

**Figure 4 sensors-21-02438-f004:**
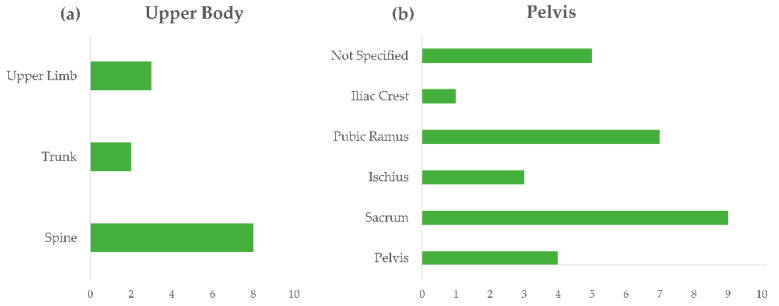
Reviewed articles regarding location of stress fractures in the upper body (**a**) and pelvis (**b**).

**Figure 5 sensors-21-02438-f005:**
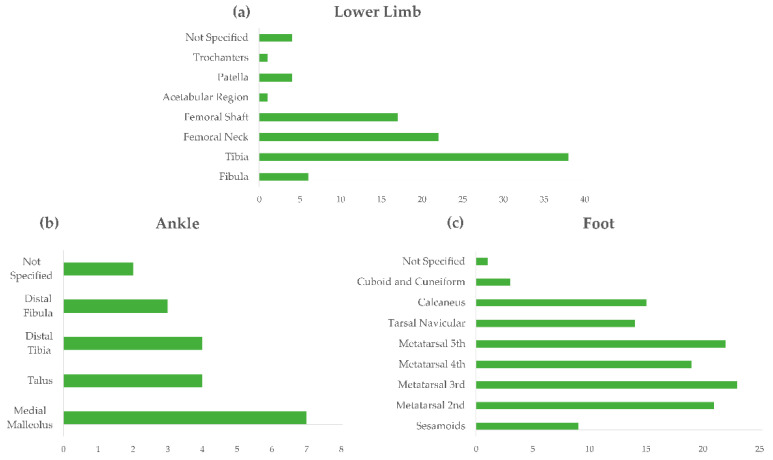
Reviewed articles regarding location of stress fractures in lower limb (**a**), ankle (**b**), and foot (**c**).

**Figure 6 sensors-21-02438-f006:**
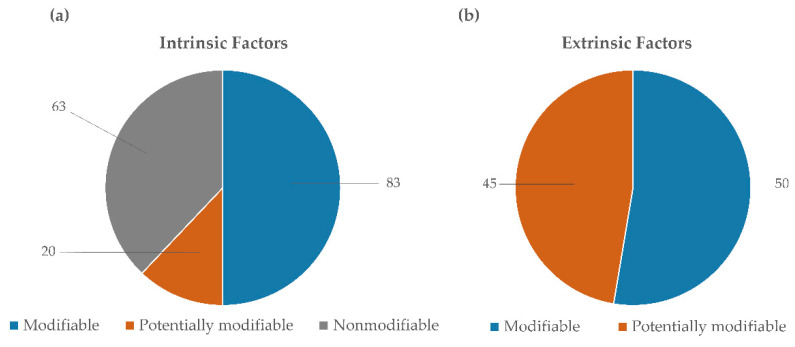
Pie charts illustrating intrinsic (**a**) and extrinsic (**b**) risk factors, and further categorization into modifiable, nonmodifiable, and potentially modifiable factors according to the number of occurrences in the articles.

**Figure 7 sensors-21-02438-f007:**
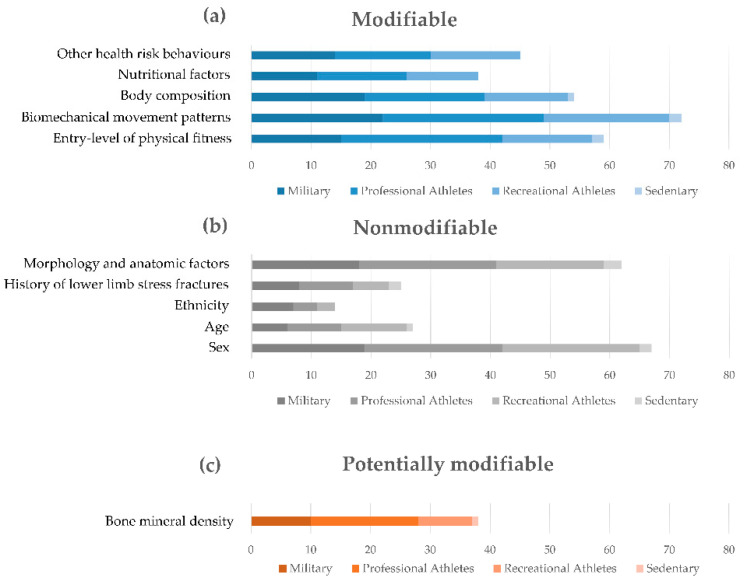
Reviewed articles regarding modifiable (**a**), nonmodifiable (**b**) and potentially modifiable (**c**) intrinsic risk factors according to the investigated population.

**Figure 8 sensors-21-02438-f008:**
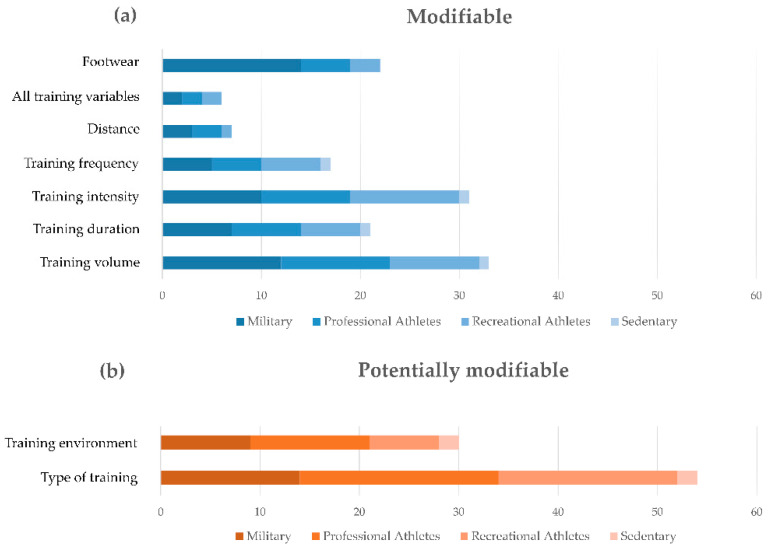
Reviewed articles regarding modifiable (**a**) and potentially modifiable (**b**) extrinsic risk factors according to the investigated population.

**Figure 9 sensors-21-02438-f009:**
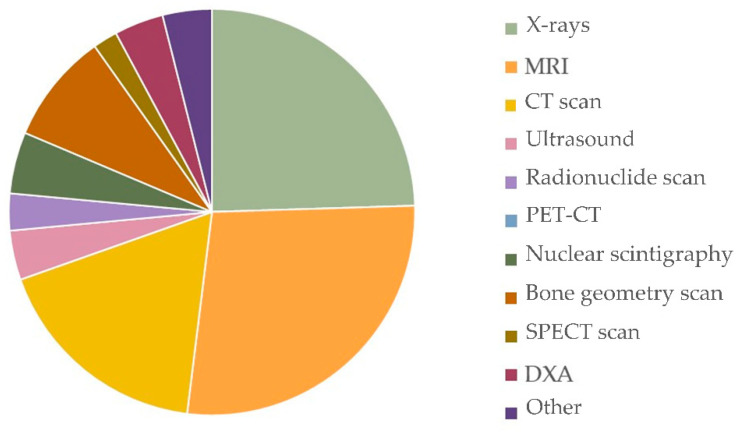
Imaging techniques.

**Figure 10 sensors-21-02438-f010:**
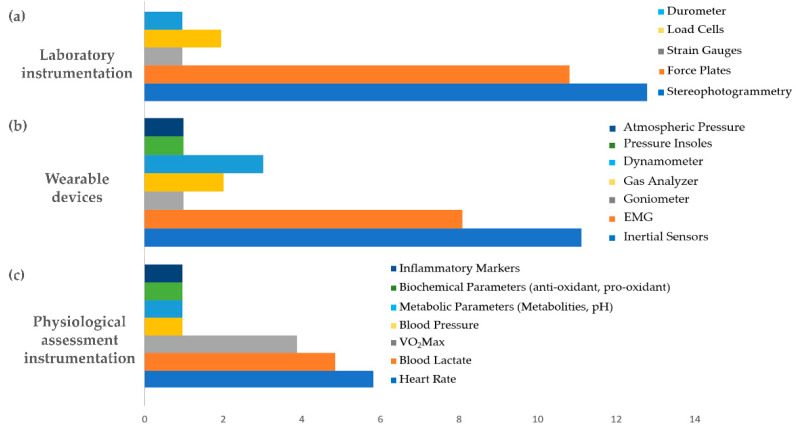
Laboratory (**a**) and wearable (**b**) instrumentation for motion analysis, and instruments for physiological assessment (**c**): occurrence of instruments and parameters in reviewed articles reporting on their use for assessing overuse-related injuries.

**Figure 11 sensors-21-02438-f011:**
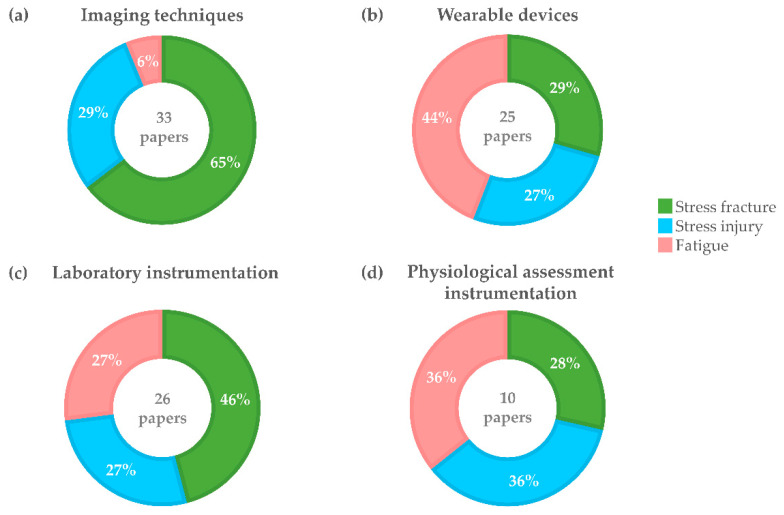
Each ring chart represents one group of instrumentation techniques, namely Imaging techniques (**a**), Wearable devices (**b**), Laboratory instrumentation (**c**) and instruments for Physiological Assessment (**d**); percentages within each ring chart refer to the distribution of stress fractures, stress injuries, and fatigue with respect to the total reported in the center.

**Figure 12 sensors-21-02438-f012:**
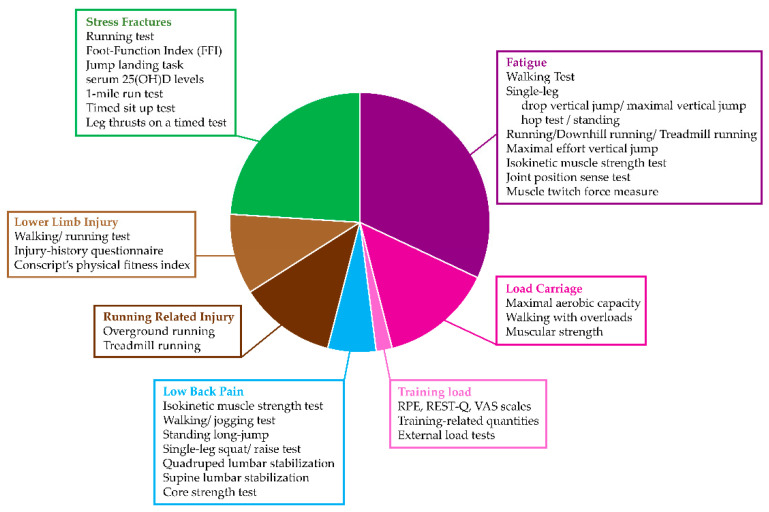
Assessment protocols associated with overuse-related injuries. Imaging-related protocols are not reported.

**Table 1 sensors-21-02438-t001:** Combined approaches of different technologies and number of articles in which they are reported.

Instrumentation	Number of Articles
Imaging techniques + Wearables	2
Imaging techniques + Laboratory	6
Imaging techniques + Physiological	0
Wearables + Laboratory	12
Wearables + Physiological	3
Laboratory + Physiological	1
Wearables + Laboratory + Physiological	1

## Data Availability

All data supporting reported results can be retrieved in the paper.

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
