# Peer review of "Overuse-Related Injuries of the Musculoskeletal System: Systematic Review and Quantitative Synthesis of Injuries, Locations, Risk Factors and Assessment Techniques"

_sensors, 2021, doi:10.3390/s21072438_

Round 1
Reviewer 1 Report
The authors provided a review of stress injuries to lower limb. Its a very well done review, however clarifying some main concerns will make this manuscript very strong
(1) Line 144 mentions MOVIDA project and it sound like a aim, however this context is not brought up at all in the introduction. The discussion first line also states this as a goal of MOVIDA project. More clarity needed on scope
(2) Line 78/79 , the authors clearly state their focus is lower limb, however the article and results also target upper limb and trunk. Perhaps the authors should have provided more discussion on the results and focus to lower limb data.
(3) A major concern is that the authors have left our a big chuck of assessment instruments used in ergonomics for assessment of repetitive/overuse stress injuries. The authors have mainly focused on sensor based, imaging and physiological approach. But, there is a whole body of instruments in ergonomics like REBA, strain index, hazard ratio, odds ratio, work cycle, etc. None of these assessments from occupational ergonomics were mentioned.
(4) Some more detail on occupational related overuse injuries and discussion on that will make the review more comprehensive. Highly recommend adding a section on that.
(5) The authors also have to define clearly the terminologies stress fracture and stress injuries.
(6) The authors also have completely missed talking about simulation and modeling efforts in this area. Most wearable sensor data and imaging data are increasingly becoming input for modeling and simulation studies. Some discussion on this is also recommended.
(7) The authors also have missed to mention about psychophysical approaches in the assessment instruments
If the authors prefer to choose only certain assessment instruments as they have done in the present version of the manuscript, it is highly recommended that they define what encompasses the definition on "instrument " with respect to the scope of this review. Further openly acknowledge that the assessment instruments in this review is not the complete list. To a large extent, except imaging, other assessment instruments listed in table 1 ( wearable senors, laboratory) are not practice gold standards. These are still in the making/development stage. In practice, mostly a combination of biomechanical, physiological, psycho-physical and psycho-social assessment instruments are used.
Highly recommend the authors to be clear on the context and acknowledge limitations of he review and what was left out unstudied in this review.
Author Response
Point-by-point reply to Reviewer 1.
The authors provided a review of stress injuries to lower limb. It’s a very well done review, however clarifying some main concerns will make this manuscript very strong
Authors. We thank the Reviewer for the positive comment. As a general action to improve the manuscript readability, a careful language revision was performed by a professional English reviewer. For detailed clarifications, a point-by-point reply is reported here below.
(1) Line 144 mentions MOVIDA project and it sound like a aim, however this context is not brought up at all in the introduction. The discussion first line also states this as a goal of MOVIDA project. More clarity needed on scope
Authors. We agree with the Reviewer. We thought confusion likely originated from a lack of clarity in the Introduction. Thus, we slightly changed the last two paragraphs of the Introduction (from line 68 of the originally submitted manuscript) as follows:
“A research project of relevance for the Army Healthcare System, which addressed open scientific questions, and involved consolidated technology and the expertise of the team of authors, was approved and funded by the Italian Ministry of Defence in 2019. Core of the MOVIDA Project (MOtor function and VItamin D: toolkit for risk Assessment and prediction) was the design, set up, validation and application of an innovative instrumental toolkit based on objective assessment instrumentation and protocols, for either indoor or outdoor use in several motor tasks, that integrates indicators of the motor function with biological, clinical, environmental and behavioural features, with the aim of extrapolating helpful information to predict the risk of musculoskeletal injury due to mechanical stress induced by overuse.
The hereby reported systematic review of the literature was thus conducted to gather information necessary to fulfil the main requirements and activities of the project. Aim of the literature search was the full characterization, wherever possible, of overuse-related injuries and muscle fatigue of the lower limbs in association with functional (biomechanical) biomarkers, in terms of type and location of injuries, populations, risk factors, suitable measurement instrumentation and protocols for functional assessment and diagnosis.”
The first sentence of the Discussion (first sentence of paragraph 4.1) was then re-written as follows:
“This systematic review focussed on the investigation of a main topic defined as “overuse-related injuries and muscle fatigue of the lower limb in association with functional (biomechanical) biomarkers: type and location of injuries, populations, risk factors, objective assessment instrumentation and protocols”.
(2) Line 78/79, the authors clearly state their focus is lower limb, however the article and results also target upper limb and trunk. Perhaps the authors should have provided more discussion on the results and focus to lower limb data.
Authors. We thank the Reviewer for this comment. Several papers dealing with lower limb overuse-related injuries also extensively reported about upper limb injuries. We thus decided to deliver a short report on these findings, too, without commenting them in detail. While we suggest to keep this additional, though partial, set of information, we brought a few changes to the manuscript so as to remove confounding parts and to highlight the relevance of lower limb findings. In detail, we:
- removed the list of body localization from the Methods (par 2.2);
- introduced a sentence at the end of par 3.3 of Results (“For the sake of clarity, it is worth to remind here that the search strategy of this review did not explicitly include upper limb overuse-related injuries; however, several papers also reported detailed information with respect to that body area, which have been reported for completeness. Of course, for that body segment, search was neither systematic nor exhaustive.”);
- modified the last period of par 4.2 of Discussion accordingly (“As clarified in the Results section, paragraph 3.3, the search strategy of this review did not explicitly include upper limb, but since overuse-related injuries to this body segment were found in some papers, they have been reported to provide a more complete picture. In any case, no interpretation is hereby given for these data.”;
- more extensively commented lower limb findings in the same 4.2 paragraph (“In particular, lower limbs were the most affected location of trauma (> 40 articles for fractures, > 30 for stress injuries, > 20 for fatigue), followed by foot (almost 30 articles for fractures and 15 for stress injuries) and spine for muscular fatigue (Figure 3). When referring to lower limbs (Figure 5, panel at the top), several studies reported stress fractures of the femoral neck and femoral shaft, but the tibia, and more precisely its medial border, resulted as the most frequently injured bone of the lower limb; the medial tibial stress syndrome (MTSS) was reported as one of the most common causes of exercise-induced lower leg pain, with incidence reaching 20% in runners and 35% in military personnel [46]. As a key element of the kinetic chain of the human locomotor system, tibia plays a connecting function between the foot-ankle complex, which directly manages the body-ground interaction, and the hip joint, which transmits upper body motion to the bottom elements of the chain. Thus, abnormalities in structure or function of either upper-tibia or below-tibia segments contribute to the tibia burden in managing repetitive loading cycles, especially during the landing and propulsion phases of gait or running. Among the MTSS risk factors with strong or moderate evidence, in fact, literature includes hip external-rotation ROM, ankle plantar-flexion ROM, and navicular drop, besides increased BMI either due to increased body weight or to external load carriage [42,46,63,75,103-104,116,121,130]. Correspondingly, the medial malleolus, which is the most distal part of the tibia, resulted as the most common location of stress fracture at the ankle level (Figure 5, panel on the left). At the foot level, metatarsals from 2nd to 5th, which are the most involved foot structures during propulsion phases and, for some runners, also act as dampers during forefoot landing, resulted as the most common sites of injury. Calcaneal and tarsal navicular stress fractures were frequently found as well (Figure 5, panel on the right), reasonably associated with the relevant action of those structures in impact absorption during foot landing.”)
Authors. Points 3 and 4 have been moved below and addressed together with points 6 and 7, please find Authors’ reply at the end of point 7.
(5) The authors also have to define clearly the terminologies stress fracture and stress injuries.
Authors. We have addressed this aspect at the beginning of the Methods, before introducing the search terms. Specifically, we clarified that: “stress fracture” refers to a fatigue-induced bone fracture caused by repeated submaximal loading; “stress injury” is a synthetic term which refers to stress-related injuries of soft tissues like muscles, tendons and ligaments following their strain under repeated mechanical stress. We also clearly stated that we are not referring to acute traumas.
The added sentence is: “Overuse-related injuries were classified either as stress fractures, referring to fatigue-induced bone fractures caused by repeated submaximal loading, or as stress injuries, the latter being a synthetic way to refer to stress-related injuries of soft tissues like muscles, tendons and ligaments following their strain under repeated mechanical stress. Neither stress fractures nor stress injuries include acute traumas.”
(3) A major concern is that the authors have left out a big chuck of assessment instruments used in ergonomics for assessment of repetitive/overuse stress injuries. The authors have mainly focused on sensor based, imaging and physiological approach. But, there is a whole body of instruments in ergonomics like REBA, strain index, hazard ratio, odds ratio, work cycle, etc. None of these assessments from occupational ergonomics were mentioned.
(4) Some more detail on occupational related overuse injuries and discussion on that will make the review more comprehensive. Highly recommend adding a section on that.
(6) The authors also have completely missed talking about simulation and modeling efforts in this area. Most wearable sensor data and imaging data are increasingly becoming input for modeling and simulation studies. Some discussion on this is also recommended.
(7) The authors also have missed to mention about psychophysical approaches in the assessment instruments
If the authors prefer to choose only certain assessment instruments as they have done in the present version of the manuscript, it is highly recommended that they define what encompasses the definition on "instrument " with respect to the scope of this review. Further openly acknowledge that the assessment instruments in this review is not the complete list. To a large extent, except imaging, other assessment instruments listed in table 1 (wearable sensors, laboratory) are not practice gold standards. These are still in the making/development stage. In practice, mostly a combination of biomechanical, physiological, psycho-physical and psycho-social assessment instruments are used.
Highly recommend the authors to be clear on the context and acknowledge limitations of the review and what was left out unstudied in this review.
Authors. We understood the Reviewer’s concerns and accepted the suggestion and the recommendation to clarify as much as possible the scope of the review and its limitations. Whenever possible, we also added small contributions to partly address some of the raised issues.
As a first action, we better clarified the research question of the review. Basically, we aimed at collecting, analysing and synthesizing evidence, or at least reliable information to characterize overuse-related damages to the musculo-skeletal system with special focus on lower limb, most frequently reported risk factors, and suitable configurations of objective assessment scenarios (namely, measurement devices, test protocols, measurable quantities or extrapolated indicators). This research question was the leading driver of the implemented literature search strategy, which thus prevented us from retrieving exhaustive information on some of the aspects mentioned by the Reviewer: as an example, we did not explicitly search the terms “occupational ergonomics” in the first-level search (review papers), neither we investigated this aspect in the second-level search (papers cited in the review papers). On the other side, our research question strongly influenced the way we analyzed and synthesized the retrieved papers. In particular, we did not go deep into: indicators and models like hazard or odds ratios and regressive or logistic models to estimate risk factors (we were only interested in retrieving the most frequently found risk factors); assessment instruments – other than objective measurement devices - typically used in clinical or occupational ergonomics examination (among which pain, discomfort or complaint scales, strain index, rapid entire body assessment score, psycho-social or psycho-physical assessment tools); occupational scenarios other than the military or the ground-related sports context; populations like developmental age individuals, seriously impaired individuals, and the elderly; upper body segments. We clarified these “limitations” in the very last period of the Conclusions, where we however explained that, whenever retrieved, we briefly summarized those findings in the review, and that although out of the specific focus of this review, the retrieved material could constitute a starting point to develop toolkits dedicated to other populations (elderly), body localization (upper-body segments) and occupational settings (other workers).
Specifically, for the occupational ergonomics issue, we added a short paragraph (paragraph 3.7) at the end of the Results, as follows:
3.7 The occupational ergonomics domain
The literature search strategy implemented in this study was not designed to include exhaustive information on other important aspects associated with the topic of overuse-related injuries as is the case for occupational ergonomics. The investigation of this specific domain was beyond the scope of the study, and the search terms “occupational ergonomics” were not explicitly used. The few findings that came out from the literature database are here shortly summarized without any claim of completeness. Briefly:
- 20 papers (9 reviews and 11 original studies) dealt with occupational ergonomics and overuse-related injuries. Specifically, 9 referred to the military field (6 reviews, 3 original studies) [67,89,113,127,135,137,170,173,223]; 1 review dealt with bricklayers and construction supervisors [224]; 4 (2 reviews, 2 original studies) more generally referred to long-standing posture at work [62,70,133,171]; 4 (original studies) addressed athletes [102,116,209,228]; 2 original papers dealt with slip-induced falls in workers [100,114].
- Of the 20 papers, 5 (3 reviews, 2 original studies) dealt with assessment instruments like strain index, rapid entire body assessment (REBA) score, pain/discomfort/complaints scales among which VAS, Borg and Likert scale [62,70,133,165,171]; 10 papers (1 review, 9 original studies) reported models and techniques (regressive/logistic models, odds ratio, hazard ratio) to detect risk factors [46,81,89,102,106,116,121,135,173,228].
With respect to the specific issue of “instruments”, we clarified it:
- in paragraph 2.4 of the Methods, which is now as follows: “Another relevant research question of the MOVIDA project and, consequently, of this literature review, concerned the investigation of the most common instruments and tools used to functionally assess stress fractures, stress injuries, and fatigue. In the case of stress fractures and stress injuries, assessment is intended as either direct – through clinical examination and/or diagnostic instrumentation – or indirect (functional), the latter based on the assessment of the impact of stress damage on musculo-skeletal performance during assigned motor tasks. As for muscle fatigue due to mechanical overload, the search focussed on functional assessment only.
Assessment may be further grouped into objective and subjective assessment, the latter including assessment tools traditionally used in occupational ergonomics (among which self-administered questionnaires, qualitative or semi-quantitative, operator-mediated scales, psycho-physical and psycho-social scales). Given the focus of the project on an innovative toolkit to perform objective measures, we limited our analysis to objective assessment, requiring measurement devices.
Four main categories of objective assessment instrumentation were identified to address the analysis of the literature, namely, imaging techniques, wearable instrumentation for motion analysis, laboratory (i.e. not wearable) devices for motion analysis, and instruments for physiological assessment.
Protocols contemplating imaging techniques were not investigated; rather, functional assessment protocols based on technology and, when retrieved, protocols to evaluate Vitamin D level or bone density were separately synthesized for stress fractures, stress injuries and fatigue.”
- in the heading of paragraph 3.5 of the Results, which is now “3.5. Objective assessment instrumentation”
- in paragraph 4.4 of the Discussion, where the first sentence is now “The literature search brought to light four main categories of objective assessment devices covering the domains of imaging techniques, wearables and laboratory devices, and instruments for physiological assessment.”
We modified the Conclusions (last period) as follows:
This systematic review focussed on collection, analysis and synthesis of evidence, or at least of reliable information to characterise overuse-related damage to the musculo-skeletal system with special focus on lower limb, most frequently reported risk factors, and suitable configurations of objective assessment scenarios (namely, measurement devices, test protocols, measurable quantities or extrapolated indicators). The implemented literature search strategy partly limited the retrieval of exhaustive information on some other relevant aspects associated with overuse-related injuries as, for example, occupational ergonomics (related findings briefly summarized in paragraph 3.7). The research question also influenced the way retrieved papers were analysed and synthesised In particular investigations did not go deep into: indicators or models to estimate risk factors, like hazard or odds ratios and regressive or logistic models; assessment instruments –other than objective measurement devices– typically used in clinical or occupational ergonomics examination; occupational scenarios other than the military or the ground-related sports context; populations like developmental age individuals, seriously impaired individuals, and the elderly; upper body segments. Whenever retrieved, however, those findings were briefly summarised and, although out of the specific focus of this review, they may represent a starting point to develop toolkits dedicated to other populations (elderly), body localization (upper-body segments) and occupational settings (other workers).

Reviewer 2 Report
This manuscripts were well written. I really enjoy reading it.
There are some minor layout and figure issues:
1. Page 14, Figure 11 has poor resolution. 2. Figure 5 is in page 8 but the caption is in page 9.
Author Response
Point-by-point reply to Reviewer 2.
Comments and Suggestions for Authors
This manuscripts were well written. I really enjoy reading it.
There are some minor layout and figure issues:
- Page 14, Figure 11 has poor resolution. 2. Figure 5 is in page 8 but the caption is in page 9.
Authors. We are grateful to the Reviewer for the very positive feedback. Figures editing will be surely improved following the Editor additional instructions.
Reviewer 3 Report
I have had the pleasure of reading the manuscript entitled “Injuries of musculoskeletal system due to overuse: systematic review and quantitative synthesis of injuries, locations, risk factors and assessment techniques”. The systematic review sets out to investigate a number of elements relevant to overuse injuries. They do succeed as such. However, I’m not convinced that this manuscript is ready for publication. I have given some comments below, so the authors may improve the manuscript. This is not an exhaustive list, as I find that there are too many issues with this manuscript to list the all.
- Many parts of the manuscript are hard to follow. Probably due to poor English, so I highly recommend that you consult a native English speaker.
- I think the objective of the paper is unclear. This has resulted in a lack of focus. I think the paper would benefit from a reduction in elements to cover.
- What were the inclusion criteria for studies to be included in the review?
- You mention many times that only review papers were included. Is this true or a mistake? If true, why would you include review papers and not original papers?
- Line 50-61. Please provide references to the literature for your statements.
- Line 62-63. This is not an acceptable way to provide references.
- Line 77-82. This is unclear. Please rephrase
- Line 88-89. Please justify only using PubMed/Medline and the time limitation for your search
- Line 298-299. What does a count of the number of times a risk factor occurs in a paper contribute with?
Author Response
Point-by-point reply to Reviewer 3.
Comments and Suggestions for Authors
I have had the pleasure of reading the manuscript entitled “Injuries of musculoskeletal system due to overuse: systematic review and quantitative synthesis of injuries, locations, risk factors and assessment techniques”. The systematic review sets out to investigate a number of elements relevant to overuse injuries. They do succeed as such. However, I’m not convinced that this manuscript is ready for publication. I have given some comments below, so the authors may improve the manuscript. This is not an exhaustive list, as I find that there are too many issues with this manuscript to list the all.
Authors. We thank the Reviewer for the comments below. We tried to exploit them so as to improve the overall quality of our manuscript.
- Many parts of the manuscript are hard to follow. Probably due to poor English, so I highly recommend that you consult a native English speaker.
Authors. The manuscript underwent an official English language revision. Hopefully, unclear aspects of our study are now better and more correctly explained.
- I think the objective of the paper is unclear. This has resulted in a lack of focus. I think the paper would benefit from a reduction in elements to cover.
Authors. We better clarified the main objective of this systematic review. Main intervention was in the Introduction, followed by corresponding peculiar adjustments in the manuscript. The end section of the Introduction is now:
“A research project of relevance for the Army Healthcare System, which addressed open scientific questions, and involved consolidated technology and the expertise of the team of authors, was approved and funded by the Italian Ministry of Defence in 2019. Core of the MOVIDA Project (MOtor function and VItamin D: toolkit for risk Assessment and prediction) was the design, set up, validation and application of an innovative instrumental toolkit based on objective assessment instrumentation and protocols, for either indoor or outdoor use in several motor tasks, that integrates indicators of the motor function with biological, clinical, environmental and behavioural features, with the aim of extrapolating helpful information to predict the risk of musculoskeletal injury due to mechanical stress induced by overuse.
The hereby reported systematic review of the literature was thus conducted to gather information necessary to fulfil the main requirements and activities of the project. Aim of the literature search was the full characterization, wherever possible, of overuse-related injuries and muscle fatigue of the lower limbs in association with functional (biomechanical) biomarkers, in terms of type and location of injuries, populations, risk factors, suitable measurement instrumentation and protocols for functional assessment and diagnosis.”
With these clarifications and the overall language editing, we hope all the elements in the manuscript will now appear as items of a more coordinated framework
- What were the inclusion criteria for studies to be included in the review?
Authors. We thank the Reviewer for the comment. We expanded and better clarified par 2.1 of the Methods, where we had previously quickly mentioned the exclusion criteria. The previous sentence (“Criteria for exclusion were established as: not in English; not relevant to the main topic of the search; full text not retrievable”) has now been re-written as follows:
“Criteria for exclusion were: not in English; not relevant to the main topic of the search; full text not retrievable. In detail, for a paper to be relevant to the main topic of the search, inclusion criteria followed in a first-level search were: focus on over-use-related injuries – either in terms of fractures or soft tissue injuries - or muscle fatigue; focus on lower limb, even though in combination with injuries at other body segments; being indeed a review, either narrative, or structured, or systematic with or without meta-analysis: this last point was mandatory since in some cases papers classified and retrieved as reviews do actually consist in other types of publications.”
For the sake of clarity, also sentence at previous lines 110-112 was completed as follows: “After removing duplicates and checking for exclusion criteria, the added set of articles followed the same review process as for the review papers. Only the inclusion criterion of being a review was obviously removed at this stage of the process.”
- You mention many times that only review papers were included. Is this true or a mistake? If true, why would you include review papers and not original papers?
Authors. According to our PRISMA scheme, our search was conducted at two levels. At the first level of search, we only included review papers, while at the second level of the search we also included original papers. Main reasons for this choice were reported at previous lines 402-411; however, we apologize since we found some typos in the period which might have contributed to poor clarity, Here below is the period after typos removal and language editing:
“Search terms were of course quite recurrent in the scientific literature. For this reason, rather narrow inclusion criteria were defined for the search in addition to the commonly used criteria (papers in English, no grey literature, full text availability), which are hereby briefly motivated:
- Only review papers included: systematic reviews, besides a comprehensive, critical and updated analysis of the state of the art on a research question, contain quality procedures, either standardised or adapted or purposely defined, for the assessment of the quality and adequacy of retrieved information [227]; since focus on systematic reviews only would have been too restricting, the search also included re-views that were not structured as systematic.”
- Line 50-61. Please provide references to the literature for your statements.
- Line 62-63. This is not an acceptable way to provide references.
Authors. We agree with the Reviewer. References have been detailed throughout the period (previous lines 50-61) and Bibliography updated accordingly. While reviewing the period, we added the reference to a very recent paper dealing with injury prevention in disabled athletes, and removed the reference to a paper dealing with in-vitro studies of bone fractures. Both papers were not part of the systematic review database (the former not yet published at March 2020; the latter not complying with the search criteria), thus no change was required to the systematic analysis.
- Line 77-82. This is unclear. Please rephrase
Authors. It has been rephrased as follows:
“A research project of relevance for the Army Healthcare System, which addressed open scientific questions, and involved consolidated technology and the expertise of the team of authors, was approved and funded by the Italian Ministry of Defence in 2019. Core of the MOVIDA Project (MOtor function and VItamin D: toolkit for risk Assessment and prediction) was the design, set up, validation and application of an innovative instrumental toolkit based on objective assessment instrumentation and protocols, for either indoor or outdoor use in several motor tasks, that integrates indicators of the motor function with biological, clinical, environmental and behavioural features, with the aim of extrapolating helpful information to predict the risk of musculoskeletal injury due to mechanical stress induced by overuse.
The hereby reported systematic review of the literature was thus conducted to gather information necessary to fulfil the main requirements and activities of the project. Aim of the literature search was the full characterization, wherever possible, of overuse-related injuries and muscle fatigue of the lower limbs in association with functional (biomechanical) biomarkers, in terms of type and location of injuries, populations, risk factors, suitable measurement instrumentation and protocols for functional assessment and diagnosis.”
- Line 88-89. Please justify only using PubMed/Medline and the time limitation for your search
Authors. Explanations for these choices had been reported in the Discussion (paragraph 4.1, previous lines 412-421) as follows:
“- Search limited to PubMed: in agreement with the previous limitation, this criterion likely allowed to reach all scientific relevant reviews and represented an acceptable trade-off between depth of search and resources needed for search, analysis and synthesis.
- Search limited to the last decade: this criterion is a consequence of the first one, since older evidence or relevant outcomes are surely accounted for in the review papers published in the last decade. Additionally, older publications might not be informative enough about more recent technology, training techniques or clinical approaches for diagnosis and treatment of musculoskeletal injuries; last, of course, this criterion contributed to optimizing the available resources.”
With respect to the specific issue of search limited to PubMed, Authors of an interesting recent paper reported that “Inclusion of systematic reviews was higher in MEDLINE than in any other single database (mean inclusion rate 89.7%; 95% confidence interval [89.0–90.3%])” [Goossen, K., Hess, S., Lunny, C., & Pieper, D. (2020). Database combinations to retrieve systematic reviews in overviews of reviews: a methodological study. BMC medical research methodology, 20(1), 138. https://doi.org/10.1186/s12874-020-00983-3]
- Line 298-299. What does a count of the number of times a risk factor occurs in a paper contribute with?
Authors. Since rigorous evidence approach could not be applied to the retrieved information on risk factors, we hypothesized that how frequently a risk factor was mentioned within our search database might be accounted as a possible “proxy” of a relevance/attention indicator. We added this explanation at the beginning of the corresponding paragraph of the Discussion (paragraph 4.3) as follows:
“4.3 Risk factors
A rigorous evidence-based approach could not be applied to risk factors. Thus, attention was paid to their occurrences in the retrieved literature, as if how many times a risk factor had been mentioned within the search database might be accounted for as a possible “proxy” of a relevance/attention indicator.”

Round 2
Reviewer 3 Report
I have no further comments